# Traumatic Lesser Metatarsal Fractures: A Case Series and Review of the Literature

**Connor L. Zale** [1,*]**, Melanie Cusi** [2] **and Paul M. Ryan** [3]

1    Tripler Army Medical Center, Honolulu, HI 96859, USA
2    Uniformed Services University of the Health Sciences, Bethesda, MD 20814, USA; Melanie.cusi@usuhs.edu
3    Tahoe Orthopedics & Sports Medicine, South Lake Tahoe, CA 96150, USA; pryan@bartonhealth.org
*    Correspondence: connor.l.zale.mil@mail.mil

**Abstract:** Background: metatarsal fractures are a commonly encountered musculoskeletal injury. Scant literature exists to support current treatment guidelines and few studies describe the functional or occupational outcomes of patients with lesser metatarsal fractures. The purpose of this study is to describe occupational outcomes for traumatic lesser metatarsal fractures in relation to current treatment guidelines. Methods: a retrospective review of metatarsal fractures in adult military patients 18 years and older was performed. Data included: patient demographics, fracture angulation and displacement, treatment modality, associated injuries, rate of return to active duty, requirement for additional surgeries, ability to run a 2-mile physical fitness test, and presence of permanent activity limitations. Treatment guideline criteria were applied and compared with the occupational outcomes measured. Results: 38 fractures were included. The mean age was 27.2 $\pm$ 7.8 (19–48). Here, 28 fractures were initially treated non-operatively. Fractures selected for non-operative treatment had a mean displacement of 1.7 +/− 1.1 mm and a mean angulation of 3.3 +/− 3.5° at initial presentation. Fractures selected for operative treatment had a mean displacement of 4.5 +/− 2.4 mm and a mean angulation of 15.7 +/− 13.8° at initial presentation. The return to run rate was 89% in non-operatively treated patients and 50% in operatively treated patients ($p$ = 0.02). Non-operative patients returned to running at a mean of 119 +/− 103 days and operative patients returned to running at a mean of 306 +/− 191 days ($p$ = 0.0039). 50% of operatively treated patients and 11% of non-operatively treated patents were unable to remain in the military due to their metatarsal fractures. Conclusions: patients treated non-operatively were more likely to return to running and returned to running sooner than operatively treated patients. Current treatment guidelines could not be supported or refuted based upon the study results. The occupational and functional outcomes demonstrated in this study may assist surgeons in counseling patients on their planned treatment and anticipated recovery following a lesser metatarsal fracture.

**Keywords:** metatarsal; military; lesser metatarsal; return to duty; return to run

## 1. Introduction

Metatarsal fractures are a commonly encountered musculoskeletal condition and account for 35% of all foot fractures [1]. De Boer et al. determined the average healthcare cost for all foot fracture was 836 euros for outpatient management and 6088 euros for inpatient management [2]. Metatarsals allow the distribution of a person's body weight during the stance phase. Fracture displacement can result in unbalanced forces and metatarsalgia while walking. Metatarsals have strong ligamentous attachments to each other at the base. For fracture displacement to occur, there has to be significant soft tissue damage to the intermetatarsal ligaments. Whether presenting in isolation or as a component of a more complex injury mechanism, the recommended treatment is dependent upon the independent fracture patten encountered [3]. Indications for treatment are based upon expert opinion alone and include recommendations for operative management for fractures

with greater than 10 degrees of angulation in the sagittal plane or 3–4 mm of displacement in any plane [3–7].

There is a paucity of evidence to support these guidelines. Natural history and comparative studies have not been performed. The active-duty military population is an ideal population to evaluate the efficacy of these guidelines as military service members have graded physical fitness tests documented twice a year. A running portion of the physical fitness test is included in all branches. There have been no previous studies evaluating the ability of patients with lesser metatarsal fractures to return to running. It is unclear if surgical management of displaced or minimally displaced metatarsal fracture have similar outcomes to non-operatively treated patients. The purpose of this study is to report the surgical and non-surgical outcomes for traumatic metatarsal fractures and to evaluate the efficacy of current management guidelines.

## 2. Materials and Methods

All lesser metatarsal fractures in military adult patients 18 years and older from a single institution from 2010–2019 were included in the initial analysis. Lisfranc injuries, Jones fractures, base of the 5th avulsion injuries, first metatarsal fractures and stress fractures were excluded. Family members, retirees and military veterans were excluded. Patient age, gender, side injured, treatment type (surgical and non-operative), associated injuries, date of injury, time immobilized, type of immobilization, ability to return to running and full active military duty status, displacement and angulation on radiographs were recorded from the electronic medical record. Angulation was measured on a lateral radiograph from the anatomic axis of the proximal fragment to the anatomic axis of the distal fragment. Displacement was measured with the maximally displaced metatarsal fracture. The distance from the outer cortex of the proximal fragment was measured to the outer cortex of the distal fragment. Non-operatively treated cases varied from CAM boot immobilization, casting, hard sole shoe or no immobilization. Operative cases were treated with k-wire fixation or open reduction and internal fixation. A post hoc protocol for operative indications was applied to both operative and non-operative cases retrospectively; pre-operative indications for surgery were not prospectively recorded. Statistical analysis was performed using R 3.6.0 statistical software. Chi square analysis and t-test were used to compare outcomes of operative and non-operative metatarsal fractures. A literature search was performed on PubMed on December 2021 using the terms "metatarsal" and "fracture" from 1990 to 2021. Jones and Lisfranc fractures were excluded. Articles were excluded if they included proximal 5th metatarsal fractures, non-human, biomechanic or anatomic studies. Articles were reviewed for patient outcomes and report radiographic measurements for displacement and angulation with lesser metatarsal fractures.

## 3. Results

In this case, 80 metatarsal fractures were included in the initial evaluation. As well, 20 fractures occurred in family members of a service member and were excluded. Here, 11 fractures occurred in retirees or military veterans and were excluded. Three stress fractures were excluded. One fracture was noted to be a pathologic fracture and was excluded. One Jones fracture and 2 Lisfranc fractures were excluded. Four fractures were lost to follow-up.

### 3.1. Demographics

Following exclusion, 38 fractures were included in this study. Five cases were female and 33 cases were male. Here, 18 involved the right foot and 20 injured the left foot. The mean age of the patients was $27.2 \pm 7.8$ years old (19–48).

### 3.2. Non-Operative Cases

In this case, 28 fractures were initially treated non-operatively. One non-operative case developed a non-union and was converted to operative management with a conversion rate of 3.6%. Patients were immobilized for $50.5 \pm 25.2$ days (12–109). Here, 27 cases treated

non-operatively primarily. In this case, 13 were treated with a CAM boot. Ten were treated with a hard sole shoe. Three were placed in a short leg cast. One used normal shoe wear.

*3.3. Operative Cases*

Ten fractures received operative management at initial evaluation. Of the 10 operatively treated fractures, 4 were classified as open fractures (Table 1). Three of the open fractures underwent operative fixation primarily. One open fracture was treated non-operatively but required delayed fixation due to non-union. (Figures 1 and 2) One operative fracture underwent revision surgery for a non-union with a 10% reoperation rate. Five were treated with k-wire pinning. Five were treated with open reduction and internal fixation.

**Table 1.** Fracture Treatment Type, Radiographic Measurements and Patient Outcome.

| Treatment | Fracture Type | Gender | Age | Displacement | Angulation | Treatment Type | Able to Run | Medical Board |
|---|---|---|---|---|---|---|---|---|
| Op | Open | F | 19 | 2.6 | 16 | ORIF | N | Y |
| Op | Closed | M | 22 | 2.8 | 46 | CRPP | N | Y |
| Op | Closed | F | 34 | 3.2 | 7 | CRPP | Y | N |
| Op | Closed | M | 22 | 3.2 | 17 | CRPP | N | Y |
| Op | Open | M | 25 | 3.3 | 3 | CRPP | N | Y |
| Op | Open | M | 33 | 3.8 | 28 | ORIF | Y | N |
| Op | Open | M | 19 | 4.2 | 8 | ORIF | N | NA |
| Op | Closed | M | 48 | 4.5 | 2 | ORIF | Y | N |
| Op | Open | F | 24 | 4.7 | 7 | CRPP | Y | N |
| Op | Closed | M | 28 | 6 | 24 | ORIF | Y | N |
| Op | Closed | M | 19 | 10.5 | 7 | ORIF and CRPP | N | Y |
| Non-op | Closed | M | 21 | 0 | 0 | CAM | Y | N |
| Non-op | Closed | M | 19 | 0 | 0 | HSS | Y | N |
| Non-op | Closed | M | 23 | 0 | 0 | HSS | Y | Y |
| Non-op | Closed | F | 24 | 0 | 0 | Splint and CAM | Y | N |
| Non-op | Closed | M | 36 | 0.7 | 0 | HSS | Y | N |
| Non-op | Closed | M | 31 | 1 | 0 | CAM | Y | N |
| Non-op | Closed | M | 38 | 1 | 2 | CAM | Y | N |
| Non-op | Closed | M | 25 | 1 | 8 | CAM | Y | N |
| Non-op | Closed | M | 19 | 1 | 8 | HSS | Y | N |
| Non-op | Closed | M | 40 | 1.1 | 0 | CAM | Y | N |
| Non-op | Closed | M | 23 | 1.4 | 6 | HSS | Y | N |
| Non-op | Closed | M | 26 | 1.4 | 8 | HSS | Y | N |
| Non-op | Closed | M | 31 | 1.5 | 12 | CAM and HSS | Y | N |
| Non-op | Closed | M | 22 | 1.6 | 4 | Cast and CAM | Y | N |
| Non-op | Closed | M | 39 | 1.8 | 1 | CAM | Y | N |
| Non-op | Closed | M | 23 | 1.8 | 3 | HSS | Y | N |
| Non-op | Closed | M | 26 | 1.8 | 6 | CAM | Y | N |
| Non-op | Closed | M | 27 | 1.8 | 8 | HSS | Y | N |
| Non-op | Closed | M | 37 | 1.9 | 0 | HSS | Y | N |
| Non-op | Closed | M | 20 | 2.1 | 0 | HSS | Y | N |
| Non-op | Closed | M | 21 | 2.4 | 3 | CAM | N | Y |
| Non-op | Closed | M | 26 | 2.6 | 6 | CAM | Y | N |
| Non-op | Closed | M | 19 | 2.9 | 0 | SLC | Y | N |

**Table 1.** *Cont.*

| Treatment | Fracture Type | Gender | Age | Displacement | Angulation | Treatment Type | Able to Run | Medical Board |
|-----------|---------------|--------|-----|--------------|------------|----------------|-------------|---------------|
| Non-op | Closed | M | 46 | 2.9 | 3 | Normal shoe wear | Y | N |
| Non-op | Closed | M | 20 | 3.1 | 3 | CAM | Y | N |
| Non-op | Closed | M | 30 | 3.5 | 3 | SLC | Y | N |
| Non-op | Closed | F | 27 | 3.7 | 0 | CAM | N | Y |

Gender, age, treatment type, type of fracture, initial radiographic displacement and angulation, ability to run and presence of a medical board are displayed above for all cases. Patients were placed into either a hard sole shoe (HSS), CAM walker boot, short leg cast (SLC), splint or normal shoe wear.

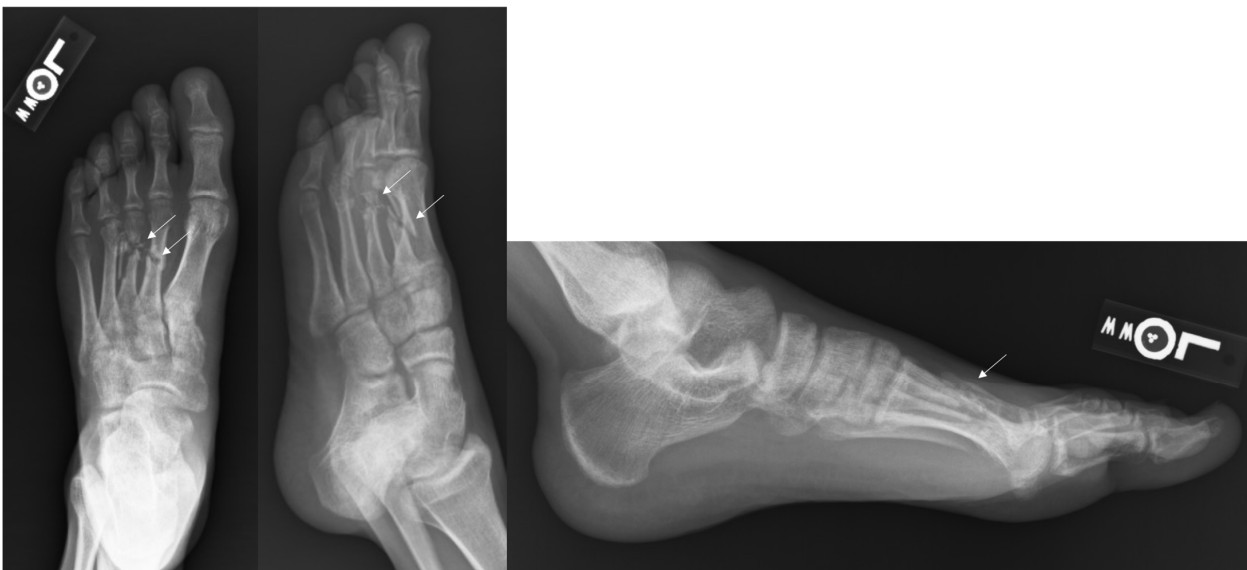

**Figure 1.** Injury Films for Displaced Metatarsal Fractures from a Gunshot. Displayed above are the injury radiographs for second and third metatarsal fractures following a gunshot injury. The arrows above identify the metatarsal fractures.

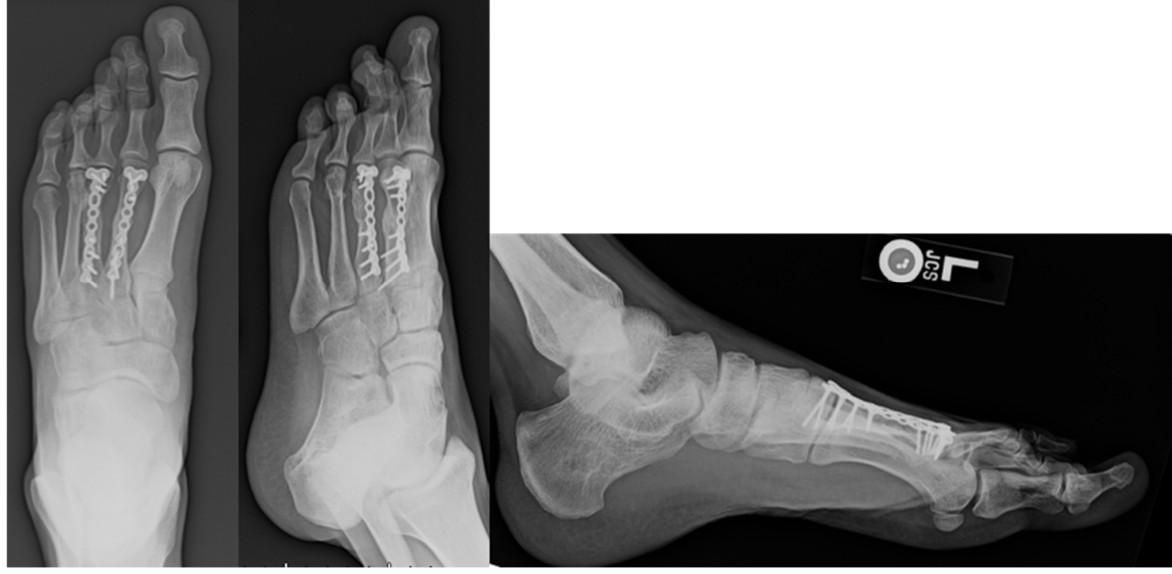

**Figure 2.** Post-operative Images for Displaced Non-union. Displayed above are the radiographs for second and third metatarsal fractures following a gunshot injury that underwent ORIF due to a non-union.

### 3.4. Radiographic Findings

A single metatarsal fracture occurred in 24 cases. Two metatarsal fractures were observed in six cases. Five cases had three metatarsal fractures. Three cases had complex forefoot and midfoot fractures. Mean displacement on injury films in any plane was $2.4 \pm 1.9$ mm (0–10.5) and the mean angulation in any plane was $6.6 \pm 9.3$ degrees (0–46). There was a statically significant difference for initial displacement ($p = 0.005$) and angulation ($p = 0.0195$) when comparing operatively and non-operatively treated cases (Table 2).

**Table 2.** Angulation and Displacement for Operative and Non-operative Cases.

|                | Non-Operative N = 28 | Operative N = 10 | *p* Value |
|----------------|----------------------|------------------|-----------|
| Open           | 1 (3.6%)             | 4 (40%)          | 0.0122    |
| Displacement   | $1.7 \pm 1.1$        | $4.5 \pm 2.4$    | 0.0051    |
| Angulation     | $3.3 \pm 3.5$        | $15.7 \pm 13.8$  | 0.0195    |
| Return to Run  | 25 (89%)             | 5 (50%)          | 0.0186    |
| Medical Board  | 3 (10.7%)            | 5 (50%)          | 0.0186    |

Displacement, angulation, fracture type, rate of return to run and rate of medical board for operative and non-operative fractures are displayed above.

In order to evaluate the outcomes with respect to accepted guidelines we first evaluated those fractures with displacement between 3 and 4 mm. Utilizing that criteria, 7 cases qualified for operative treatment (Table 3) (Figures 3 and 4). Three of the 7 cases that qualified for operative management were treated non-operatively. Two of the 3 fractures treated non-operatively healed uneventfully. The third fracture in that grouping healed but the patient was unable to return to running secondary to pain at the fracture site. When reviewing the 4 patients treated operatively, 3 of the fractures healed but 1 developed a non-union. Of the 3 operatively healed fractures that healed, there was one patient who reported inability to return to running due to pain at the fracture site.

**Table 3.** Radiographic Measurements of Metatarsal Fractures.

|               | Fracture Displacement | | |
|---------------|-----------------------|--------|---------|
|               | 3–4 mm                | >4 mm  | >10 deg |
| Non-operative | 3                     | 1      | 1       |
| Operative     | 4                     | 4      | 5       |

Breakdown cases that qualified for operative fixation by radiographic measurements are displayed above. The table also presents the number of cases that were actually treated operatively and non-operatively.

If only those patients with greater than 4 mm of displacement were considered operative candidates based upon guidelines, 5 fractures would have met criteria for surgery. Four of those 5 cases were actually treated with an operation at the time of injury. The one case that was initially treated non-operatively later developed a non-union and underwent a non-union repair. It should be noted that the fracture which developed a non-union in this grouping was secondary to gunshot injury to the foot (Figures 1 and 2). The patient who developed a non-union was unable to return to run at final follow-up. Of the 4 patients initially treated operatively, 1 was unable to return to running secondary to pain at the fracture site.

Six out of 38 fractures presented with angulation greater than 10 degrees. (Figure 5) One of these 6 fractures was treated non-operatively and healed uneventfully. Three of the 5 patients treated operatively with greater than 10 degrees of initial angulation were unable to return to running and were eventually medically discharged from the military.

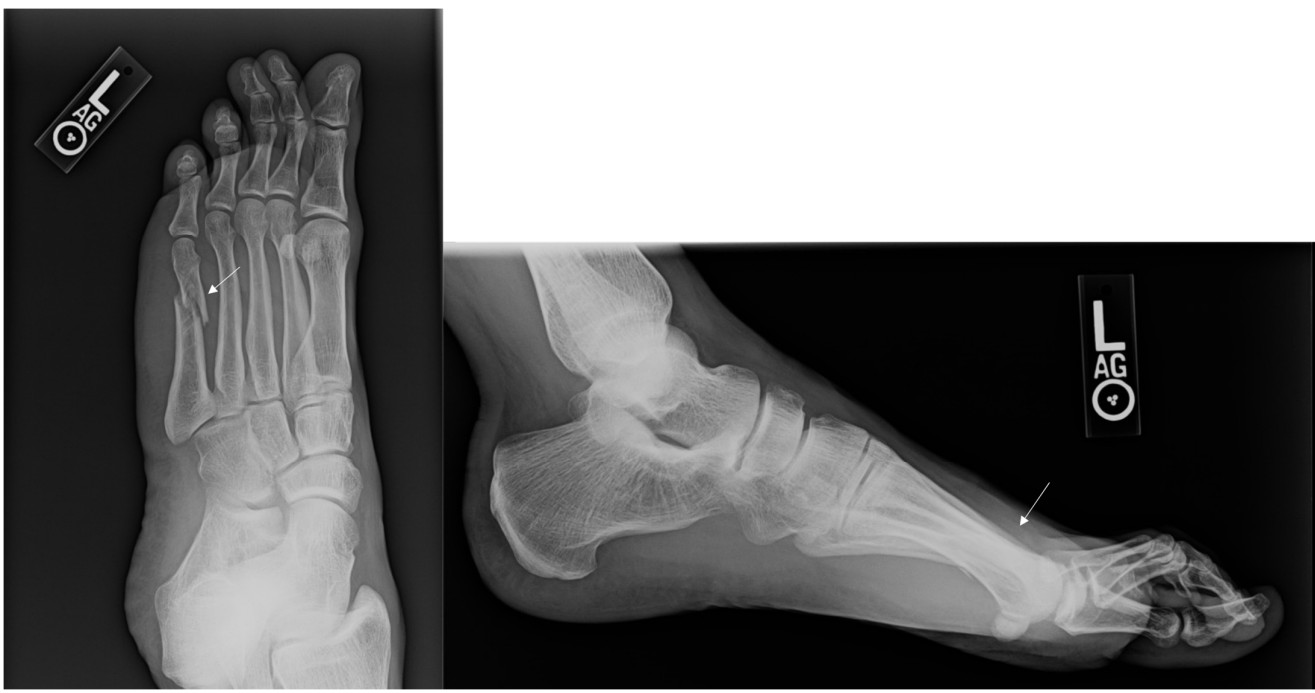

**Figure 3.** Radiographs of a Metatarsal Fracture Displaced 3–4 mm. Displayed above are radiographs of a fifth metatarsal shaft fracture that is displaced between 3–4 mm. The arrows above identify the fifth metatarsal fracture.

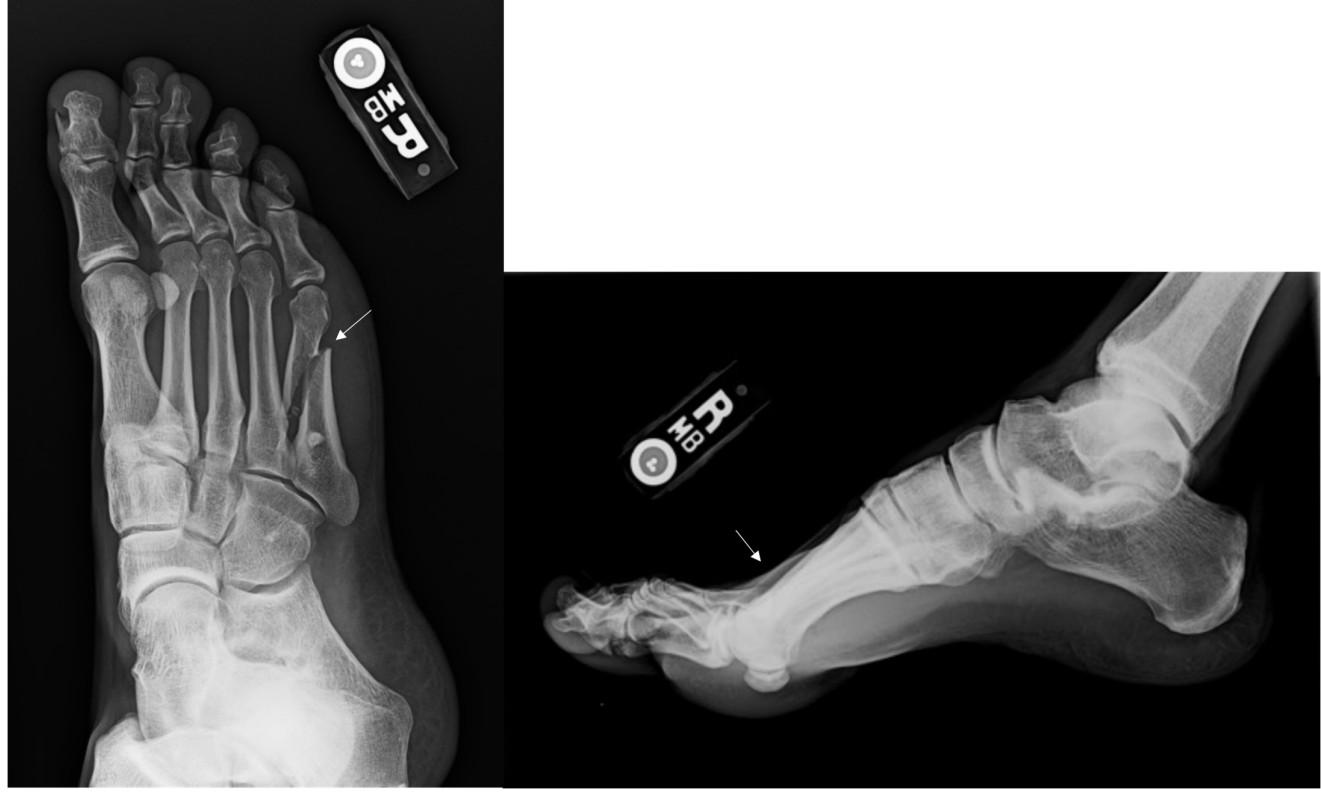

**Figure 4.** Radiographs of a Metatarsal Fracture Displaced Greater than 4 mm. Displayed above are radiographs of a fifth metatarsal shaft fracture displaced greater than 4 mm. The arrows above identify the fifth metatarsal fracture.

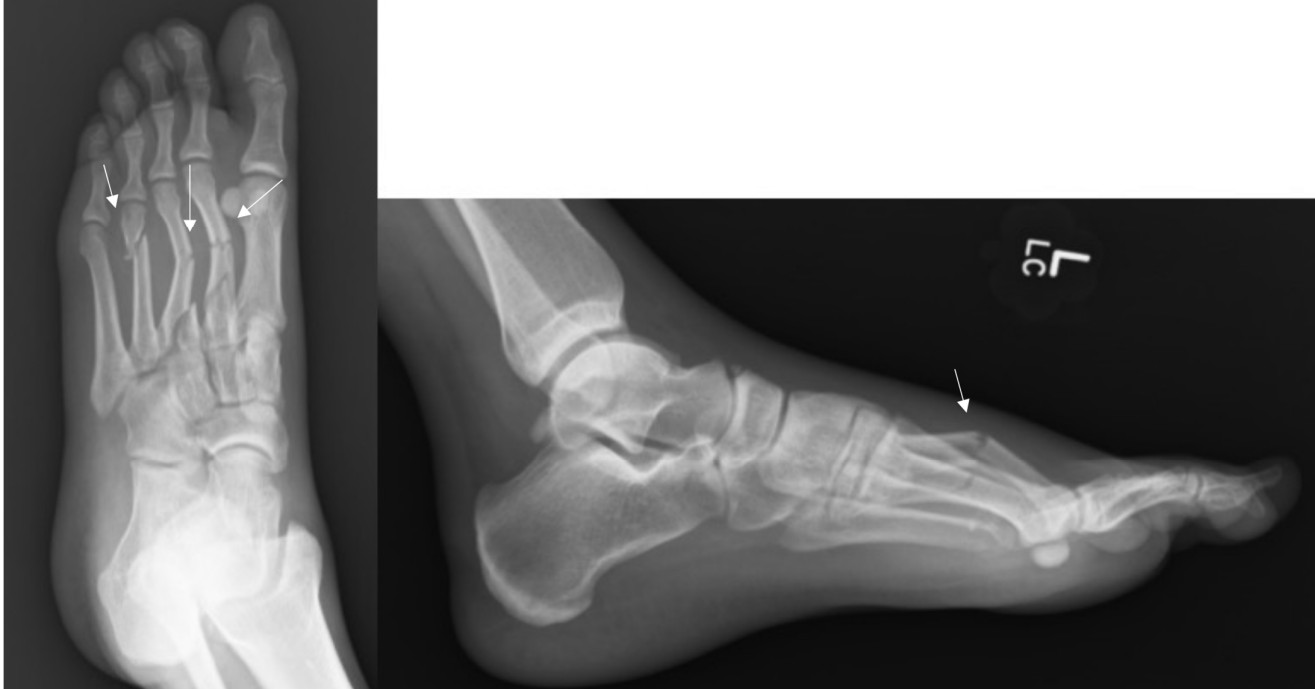

**Figure 5.** Radiographs of a Metatarsal Fracture Angulated greater than 10 degrees. Displayed above are radiographs of second, third and fourth metatarsal fractures with angulation greater than 10 degrees. The arrows above identify the central metatarsal fractures.

### 3.5. Return to Running and Military Duty

In the entire study population, 30/38 (79%) were able to return to running. Of the 8 patients who were not able to return to running, 5 (63%) were treated operatively. One of the 8 patients that were unable to return to running was initially treated non-operatively and converted to operative treatment due to development of a non-union. In total, 8 patients with metatarsal fractures were medically discharged from military duty. Five of the patients discharged were treated operatively. One of the 8 patients discharged was discharged for medical reasons unrelated to the fractured metatarsal. The remaining 7 patients were medically discharged from active duty due to their lower extremity injury (18%). There was a significant difference between non-operative and operative cases that were medically discharged from active duty ($p$ = 0.0186). There was a significant difference for operative and non-operative cases returning to running ($p$ = 0.0186) (Table 2). Operative cases returned to running with an average of 306.2 $\pm$ 190.5 days. Non-operative cases returned to running with an average of 119.3 $\pm$ 103.4 days. There was a significant difference with time to return to running between operative and non-operative cases ($p$ = 0.0039).

### 3.6. Article Review

In this case, 793 articles were initially obtained and reviewed by the study team. Additionally, 638 articles were excluded as they discussed proximal metatarsal fractures, Lisfranc injuries, non-human fractures, biomechanical and anatomic studies. Of the remaining 104 articles, 86 were excluded as they were case reports, small case series, technique articles, or review articles. Full text was obtained and reviewed for 19 remaining articles (Figure 6).

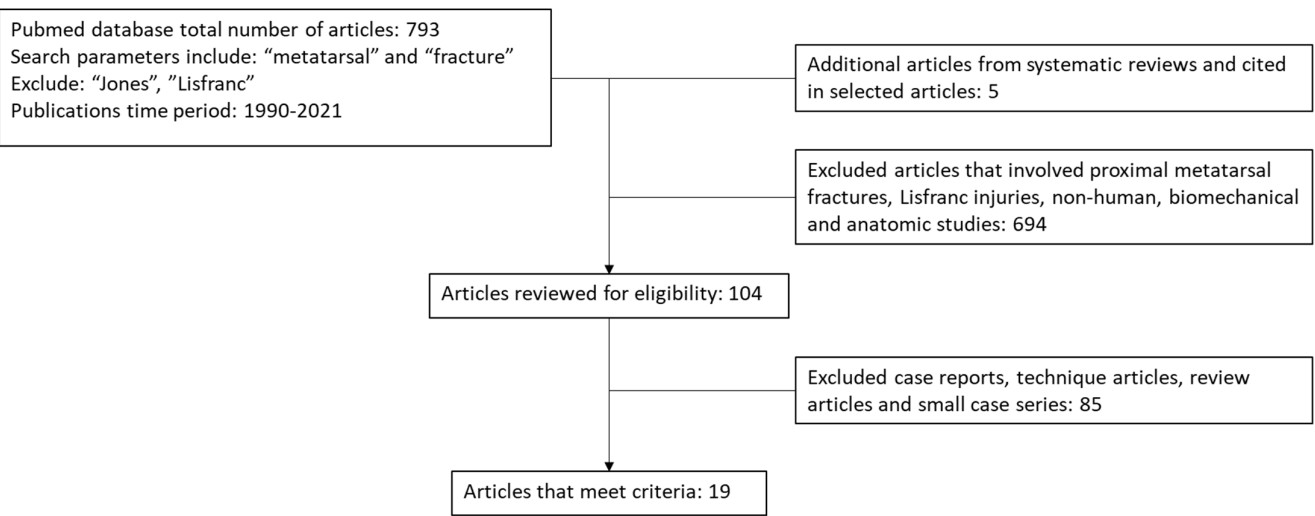

**Figure 6.** Article Search Breakdown. Database search for metatarsal fractures is displayed above. 19 articles met criteria for review of lesser metatarsal fractures.

## 4. Discussion

There is scant existing literature evaluating the operative indications for metatarsal fractures. 19 articles were obtained that reported the outcomes for lesser metatarsal fractures. One article reported a case series of open fractures treated operatively [8]. Seven articles did not report indications for treatment [9–15]. Two articles did not report mean displacement [7,16]. Nine articles referenced level 5 evidence for indications of surgery [4,5,17–23].

Non-operative management has been recommended by multiple sources for the treatment of metatarsal fractures. Aynardi et al. demonstrated successful treatment for 139/141 displaced diaphyseal fifth metatarsal fractures treated non-operatively. The reported FAAM activities daily living score average was 95.5 ± 5.7 out of 100. Two cases developed a non-union and underwent revision ORIF for a revision rate of 1.4%. The mean age in this study was 55.2 years old with a range of 16.9–95.6. In addition to including patients older than the patients in the current study, the authors did not describe outcomes in terms of occupational or recreational activities. The current study had a higher conversion rate of 3.6% to operative management.

Surgical management has been supported by other authors in high level athletes. Operative management of central metatarsals has been recommended for multiple metatarsal fractures, fractures with greater than 10 degrees of angulation in the sagittal placement or 3–4 mm of displacement in any plane [3]. Thompson et al. recommended surgical management of minimally displaced 5th metatarsal diaphyseal fractures to reduce time to full activity and restore length which was felt to prevent transfer metatarsalgia [5]. Shereff reports that displacement in the sagittal plane is not well tolerated by young active patients [4]. Goulart al recommend surgical management for fifth diaphyseal fractures in elite dancers [24]. Conversely, O'Malley et al. had success with treating fifth metatarsal diaphyseal fractures non-operatively in elite dancers [22].

Recommendations for operative or non-operative management rely on level 5 evidence. Our study is the first to report the outcomes of a military population with metatarsal fractures treated operatively and non-operatively. Our study reports the mean displacement for operative and non-operative cases and reviews the occupational outcomes of patients in regards to initial displacement and treatment. Displacement and angulation cut-offs for surgery may need further evaluation. Two of the 3 cases with displacement between 3 and 4 mm treated non-operatively were able to return to full duty and running. The single case treated non-operatively with >4 mm displacement was unable to return to running. The

single case with angular displacement greater than 10 degrees treated non-operatively was able to return to full activity.

In the current study, the overall rate of returning to running was 89% among non-operative patients and 50% for operatively treated patients. The medical discharge rate was 50% for operative cases and 10.7% for non-operative cases. Non-operative cases returned to running sooner than operative cases. Current treatment guidelines are not evidenced based and we were not able to either support or refute the guidelines with the numbers available in our study. Metatarsal fractures remain heterogenous injuries that have varied clinical outcomes in an active population. Further studies will need to be conducted to determine ideal guidelines for management of central metatarsal fractures.

This study has limitations. The sample size was small. Results are descriptive rather than comparative as the two cohorts were differentiated based upon treatment received rather than on fracture morphology. Both the operative and non-operative treatments were not standardized. Operative indications were not recorded pre-operatively. With the numbers available, we were not able to provide sub-group analysis in relation to the various operative procedures performed and rather grouped all operative procedures together in one cohort. We were unable to determine sub-group analysis by immobilization method due to lack of standardization and number of patients using multiple immobilization methods. The occupational and functional outcomes described were in a military population and may not be applicable to other patient groups.

## 5. Conclusions

The overall rate of returning to running was 89% among non-operative patients and 50% for operatively treated patients. Non-operatively treated patients returned to running sooner than those patients who received an operation. Although the data presented in this study did not confirm or refute current treatment guidelines, the occupational outcomes presented in this study may help surgeons and patients in their shared decision-making process.

**Author Contributions:** Conceptualization, C.L.Z. and P.M.R.; methodology, C.L.Z. and P.M.R.; software, C.L.Z.; investigation, C.L.Z. and M.C.; data curation, C.L.Z. and P.M.R.; writing—original draft preparation, C.L.Z. and M.C.; writing—review and editing, C.L.Z., M.C. and P.M.R.; supervision, P.M.R.; project administration, P.M.R. All authors have read and agreed to the published version of the manuscript.

**Funding:** This research received no external funding.

**Institutional Review Board Statement:** The study was conducted in accordance with the Declaration of Helsinki, and approved by the Institutional Review Board of Tripler Army Medical Center (21R07, 06MAY2021) for studies involving humans.

**Informed Consent Statement:** Patient consent was waived due to this study being retrospective in nature and approved by institutional IRB.

**Data Availability Statement:** Not applicable.

**Acknowledgments:** There are no additional acknowledgements for this manuscript.

**Conflicts of Interest:** There are no disclosures or conflicts of interest from the authors.

**Disclaimer:** The views expressed in this abstract/manuscript are those of the author(s) and do not reflect the official policy or position of the Uniformed Services University of the Health Sciences, Department of the Army, Department of Defense, or the US Government.

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
