# Peer review of "Traumatic Lesser Metatarsal Fractures: A Case Series and Review of the Literature"

_2673-4036, doi:10.3390/osteology2020009_

Round 1

Reviewer 1 Report

Comments and Suggestions for Authors

please see attached word file

Author Response

The authors would like to thank the reviewer for their time and comments.  We have made changes as requested.  If the change was not made, the comments were addressed in the attached document.  Thank you.

Reviewer 2 Report

Comments and Suggestions for Authors

Dear Editor,

dear authors,

Thank you very much for giving me the opportunity to review the manuscript entitled:

 Traumatic Lesser Metatarsal Fractures in a Military Population

The Purpose of this study was to assess outcomes for traumatic metatarsal fractures in a military population according to fracture patterns and treatment methods.

Although, the manuscript covers an interesting topic the manuscript shows several several severe flaws according writing, methodology and technical issues.

Introduction

It remains unclear why authors choose a military population for their study. Further, “military population” need better definition. You could point out the special needs and characteristics of your patients and why that problem is most important in this field or different to other previous studies. You should also provide details about the overall aims restoring native foot anatomy and give some biomechanical background.

Materials & Methods

  • What´s the overall population your patients were recruited from?
  • The in- and exclusion criteria belong to the M&M section
  • There is no information about the diagnostic workup or the time from trauma to diagnosis. Further, you should describe every variable you evaluated in detail
  • There is no information about the details of a surgical or conservative treatment protocol, and no rebab details
  • Absolutely no information about statistic methods used (test variables, software…). How did you calculate the p-values?
  • Information about the classification of fracture would be helpful. Did you use for example the AO classification on MT fractures? You should also classify the open fractures..for example Anderson/Gustilo
  • It remains unclear why fractures were treated initially in the one or another way. Did you treat open fractures conservatively? Indication for surgery not only depends on the amount of dislocation but also on the fracture pattern / anatomic location..therefore, a more detailed description is necessary
  • The overall amout of cases is low for the statistical methods used
  • How did you measure displacement and angulation? How did you measure 0- 0.7mm displacement? Another case shows 1mm displacement and 8 degree angulation…
  • What bones were exactly affected? Obviously you also included multiple fractures..

Discussion

Is not adequate for reflecting your results. Reads more like a review of literature. Consider also inserting a limitations section.

Author Response

The authors would like to thank the reviewer for their time and comments.  We have made changes.  If the change was not made, the comments were addressed in the attached document.  Thank you.

Reviewer 3 Report

Comments and Suggestions for Authors

Dear Authors,

Congratulations for the work done. This is a well-developed research of interest to the scientific community. However, the manuscript presents formal errors that should be corrected before possible publication.

ABSTRACT: Correct and adequate.

INTRODUCTION: This section should be expanded to adequately present the clinical and socio-health impact of the problem to be solved with this research.

MATERIAL AND METHODS: If the research intends to review the literature (as stated in its title), the methodology should be thoroughly expanded and corrected.

RESULTS: It happens the same as in the previous section.

DISCUSSION: Taking into account the corrections to be made in the two previous sections, I cannot confirm the adequacy or otherwise of this section.

Please, when authors submit a manuscript for evaluation they should make sure that the version they submit is the final version and that the change control is no longer active. It is very unfortunate that the authors have made a mistake.

Kind regards

Author Response

Thank you for your suggestions.  We have updated the methods and results to include the literature review.  We have also included the economic cost in the introduction.  We look forward to your updated review.

Round 2

Reviewer 1 Report

Comments and Suggestions for Authors

Significant improvement has been made to the manuscript by Zale et al. A few comments:

  • Since there are different methods of non-operative and operative treatment, those methods should be added to table 1
  • Was there any difference in outcome among the different types of non-operative and operative treatments? For instance in return to running or rate of medical discharge?
  • For the pictures of fractures, please point out where the fractures are for readers

Author Response

Since there are different methods of non-operative and operative treatment, those methods should be added to table 1

We have expanded table 1 information as requested above.

Was there any difference in outcome among the different types of non-operative and operative treatments? For instance in return to running or rate of medical discharge?

We were unable to perform subgroup analysis due to patients receiving multiple types of immobilization or mixed fixation constructs.  We discuss this in line 207-209 in discussion.

For the pictures of fractures, please point out where the fractures are for readers

We have added arrows to the images.  Thank you for reviewing our manuscript.

Reviewer 2 Report

Comments and Suggestions for Authors

Dear Editor,

dear authors,

Thank you very much for the changes made to your manuscript. Despite the fact, that some parts of the manuscript have improved I still observe severe technical issues in presenting methodology and results. E.g. the Discussion part ist is inadequate and does not discuss/reflect your own results properly. Moreover, the most serious issue I have is the missing clear indications / treatment protocols for operative vs non-operative cases in your small group of patients.

Author Response

Thank you for your response to our article.  We have expanded the discussion section and we have reflected our results in lines 186-203.  We also discuss that the operative indications were not recorded in lines 207-212.

Reviewer 3 Report

Comments and Suggestions for Authors

Dear Authors,

Congratulations for the work done.

The suggested recommendations have been followed and the manuscript has improved significantly in quality.

Kind regards.

Round 3

Reviewer 2 Report

Comments and Suggestions for Authors

Dear Editor,

dear authors,

although the authors have improved their manuscript adding some more details, I still have the feeling that due to methodological flaws in the study design the results do not allow any further insights in the treatment of metatarsal fractures.

Further presentation of results and discussion still demonstrate technical flaws in writing.

Author Response

Thank you for raising concern.  We have reviewed the manuscript to ensure grammar and spelling is correct.

The guidelines for treatment of metatarsal fractures is based upon level V evidence (expert opinion).  This is the first study to evaluate these guidelines as they relate to clinical outcomes.  While this study on its own does not independently support or refute the guidelines, it is a valuable addition to the medical literature.
